



# Novel approach to estimate the water isotope diffusion length in deep ice cores with an application to MIS 19 in the EPICA Dome C ice core

Fyntan Shaw[1], Andrew M. Dolman[1], Torben Kunz[1], Vasileios Gkinis[2], and Thomas Laepple[1, 3]

[1]Alfred Wegener Institute, Helmholtz Centre for Polar and Marine Research, Potsdam, Germany
[2]Niels Bohr Institute, Physics of Ice, Climate, and Earth, University of Copenhagen, Copenhagen, Denmark
[3]University of Bremen, MARUM - Centre for Marine Environmental Sciences and Faculty of Geosciences, Bremen, Germany

**Correspondence:** Fyntan Shaw (fyntan.shaw@awi.de)

**Abstract.** Accurate estimates of water isotope diffusion lengths are crucial when reconstructing and interpreting water isotope records from ice cores. This is especially true in the deepest, oldest sections of deep ice cores, where thermally enhanced diffusive processes have acted over millennia on extremely thinned ice. Previous estimation methods, used with great success in shallower, younger ice cores, falter when applied to these deep sections, as they fail to account for the statistics of the climate on millennial timescales. Here, we present a new method to estimate the diffusion length and apply it to the Marine Isotope Stage 19 (MIS 19) interglacial at the bottom of the EPICA Dome C (EDC) ice core. In contrast to the conventional estimator, our method uses other interglacial periods taken from further up in the ice core to estimate the structure of the variability before diffusion. Through use of a Bayesian framework, we are able to constrain our fit while propagating the uncertainty in our assumptions. We estimate a diffusion length of $31 \pm 5$ cm for the MIS 19 period, which is significantly smaller than previously estimated (40 cm - 60 cm). Similar results were obtained for each interglacial used to represent the undiffused climate signal, demonstrating the robustness of our estimate. Our result suggests better preservation of the climate signal at the bottom of EDC and likely other deep ice cores, offering greater potentially recoverable temporal resolution and improved reconstructions through deconvolution.

## 1 Introduction

Large ice sheets from the polar regions offer unique insights into the climate up to hundreds of thousands of years ago. The drilling of deep ice cores in Greenland and Antarctica enables measurements of water isotopic ratios ($\delta^{18}O$, $\delta D$, $\delta^{17}O$) impacted by fractionation effects upon evaporation and condensation. These ratios have been shown to relate to atmospheric temperatures at the time of deposition as snowfall (Dansgaard, 1964) and therefore provide a valuable proxy of past climate conditions. However, water isotope records are not perfectly preserved, partially due to the molecules dispersing over time, smoothing the profile by attenuating high frequency variability. This displacement is known as diffusion and occurs both in the firn, due to snow-vapour exchange in the pores (Johnsen, 1977; Whillans and Grootes, 1985), and in ice through processes such as ice diffusivity (Ramseier, 1967) and liquid water veins (Nye, 1998). Diffusion in ice is a much slower process than





in firn, but can act over much longer time periods, with the oldest, deepest sections of deep ice cores most affected. Since ice diffusion increases with temperature, the warming of deep ice due to geothermal heat from the bedrock further accelerates the
process. Additionally, the effect is exacerbated on the temporal scale by extreme layer thinning from ice flow. Collectively, these conditions can result in the attenuation of variability up to millennial timescales in deep ice cores (Pol et al., 2010).

Water isotope diffusion can be characterised by the diffusion length, defined as the average displacement of water molecules along the vertical axis relative to their initial position within the ice sheet. In addition to informing which frequencies climate
variability is preserved, knowledge of the diffusion length can enable techniques used to recover some of the lost information, such as deconvolution (Johnsen, 1977). These reconstructions are extremely sensitive to the diffusion length, so obtaining accurate values is crucial for interpreting the isotopic data. Furthermore, the temperature dependence of the diffusion process reflected on the magnitude of the diffusion length constitutes the latter as a good candidate for a firn temperature proxy (Simonsen et al., 2011; Gkinis et al., 2014; Van Der Wel et al., 2015).


By comparing the power spectrum of a diffused water isotope record with that of the undiffused isotopic climate signal it is possible to estimate the diffusion length. Current estimation methods assume that the isotopic variability before diffusion is constant across all frequencies, i.e. white-noise (Johnsen et al., 2000; Gkinis et al., 2014; Jones et al., 2017; Kahle et al., 2018; Holme et al., 2018). This assumption is justified by the strong noise generated by precipitation intermittency and the
stratigraphic noise, which dominates the signal up to decadal or multi-centennial timescales depending on the accumulation conditions of the site (Johnsen et al., 2000; Casado et al., 2020). However, on the longer timescales observed at the bottom of deep ice cores, the isotopic profile is not accurately represented by white-noise. This is evident from other ice cores, which demonstrate strong variability over millennia, such as Dansgaard-Oeschgar events from Greenland cores (NGRIP members, 2004; NEEM community members, 2013), or Antarctic Isotope Maxima events visible in Antarctic cores (Petit et al., 1999;
EPICA community members, 2004). To accurately represent the statistical properties of such millennial variability, the estimation method requires a modified approach.

In this study we estimate the diffusion length for the Marine Isotope Stage (MIS) 19 section of the EPICA Dome Concordia (EDC) ice core (763 - 795 ka). Previously estimations suggest a diffusion length between 40 cm and 60 cm for the MIS 19
interglacial (Pol et al., 2010). In contrast, diffusion models using the physical properties of the ice core predict values between 16 cm and 22 cm (Pol et al., 2010; Grisart et al., 2022), which poses the question whether the model physics is wrong, and/or the statistical estimator is biased when applied to deep ice. We address the latter point, removing the assumption that the climate signal can be approximated by white-noise. Instead, the initial climate is inferred from similar periods further up in the core where the water isotopes have not undergone significant ice diffusion. We use a Bayesian methodology that propagates
uncertainty in this reference climate spectrum while also constraining parameters to physically realistic ranges. This paper explains the new approach and discusses possible future applications and improvements. The new diffusion length estimate is



compared with previous estimations and will serve as an indicator for how significant deep ice diffusion will be in future deep ice cores such as the Beyond EPICA - Oldest Ice Core (BE-OIC).

## 2 Data and Methods

### 2.1 Water isotope data

We use discrete $\delta^{18}$O data from the EDC ice core, measured at the Niels Bohr Institute, University of Copenhagen with a water-CO$_2$ equilibration mass spectrometry system (Finnegan MAT 251) (Grisart et al., 2022), https://doi.org/10.1594/PANGAEA.939445. The data has a resolution of 11 cm, a reported accuracy of 0.07‰ and was dated using the Antarctic ice-core chronology (AICC 2012) (Veres et al., 2013). For the diffusion length estimate, we define MIS 19 from a depth of 3147 m (748 ka) to 3190 m (802 ka), which marks the beginning of the MIS 18 glacial period and the termination of the MIS 20 glacial period respectively (Pol et al., 2010). Depth ranges for the more recent interglacial periods MIS 1 (the Holocene), MIS 5 and MIS 9 are given in Table 1, along with depth ranges where data is missing. All section selections are a trade-off between using a long section allowing for a more precise statistical estimation and using a shorter section only containing warm periods.

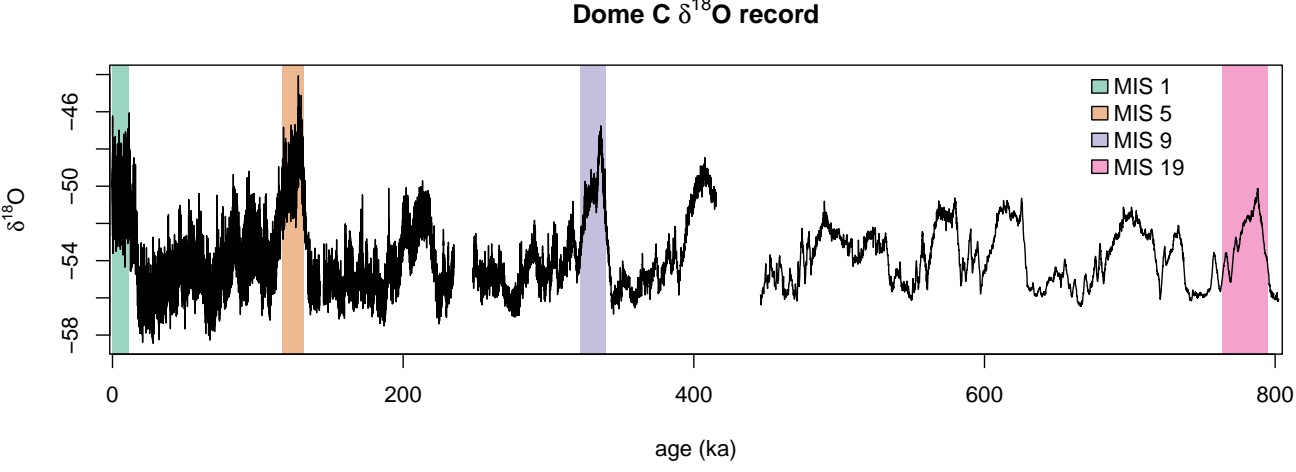

**Figure 1.** Full high resolution EDC $\delta^{18}$O record (Grisart et al., 2022), with selected interglacial windows used in this study highlighted.



**Table 1.** Depth ranges of analysed data and the corresponding time periods. Also included are the depths and times at which the $\delta^{18}$O data was missing. MIS 9 and MIS 19 contained no missing values.

| Interglacial | Depth Range (m) | Time Period (ka) | Missing $\delta^{18}$O Data Depth Ranges (m) | Missing $\delta^{18}$O Data Time Periods (ka) |
| --- | --- | --- | --- | --- |
| MIS 1 | 7.755-354.145 | 0 - 11.6 | 119.955-120.615 | 3.15-3.17 |
| | | | 234.355-234.795 | 7.27-7.28 |
| | | | 288.805-290.345 | 9.32-9.38 |
| MIS 5 | 1539.505-1734.755 | 116.7-131.6 | 1583.505-1594.395 | 120.3-121.1 |
| | | | 1638.505-1649.395 | 124.5-125.4 |
| | | | 1693.505-1704.395 | 128.5-129.3 |
| MIS 9 | 2526.645-2592.645 | 321.9-339.3 | - | - |
| MIS 19 | 3159.145-3185.545 | 763.6-795.0 | - | - |

## 2.2 Method overview

In summary, the diffusion length in deep ice is estimated using a modification of the existing statistical model, by representing the power spectrum of the water isotope record before diffusion as a power law. An appropriate power law is estimated from selected water isotope data from shallower sections of the same ice core, where diffusion will not have had time to affect the relevant frequencies, spanning time periods of similar climate state and length to the deep ice section.

## 2.3 Spectral analysis

The diffusion length and the properties of the isotopic variability are estimated in the frequency domain. For this, we use the raw periodogram on linearly detrended data with a split cosine bell taper of 10% as this estimator is unbiased and we are only using it for parametric fits.

The isotope time-series are irregular in time due to ice flow thinning, which is incompatible with classical power spectra analysis. Therefore, when working with data on the time domain, the respective records were linearly interpolated to equidistant time points. Although this introduced power loss at high frequencies in the power spectra (Schulz and Stattegger, 1997; Hébert et al., 2021), our analysis did not include these frequencies and so remains largely unbiased. Gaps were only present in the temporal data used for spectral analysis, and did not exceed 100 data points (at most corresponding to 863 years).

For comparison, we also calculated diffusion lengths using the conventional white-noise model for a running window over the entire EDC ice-core record. Gaps in the data were resolved through linear interpolation, and large or frequent linear interpolated gaps can significantly effect the power spectra. Therefore, the estimate was only performed over windows in which less than 25% of the $\delta^{18}$O values were missing to minimise estimation biases, explaining the gaps seen in Fig. 5.





## 2.4 Spectral representation of diffusion length

The diffusion of water molecules averages out variability, with rapidly increasing effectiveness as frequency increases. While the lowest frequencies remain mostly unchanged, information on higher frequencies can be greatly affected. This relationship allows the diffusion length of a water isotope time-series to be estimated through analysis of its spectral properties combined with a representative model.

Mathematically, the effect of diffusion on a time-series can be represented by a convolution with a Gaussian filter (Johnsen, 1977; Johnsen et al., 2000),

$$g(z) = \frac{1}{\sigma\sqrt{2\pi}} e^{-\frac{z^2}{2\sigma^2}} \tag{1}$$

where $z$ is the depth and $\sigma$ is the diffusion length in the same units as $z$. By applying the convolution theorem, we can represent the effect of diffusion in the frequency domain with the transfer function,

$$G(k) = e^{\frac{-k^2\sigma^2}{2}} \tag{2}$$

where $k = 2\pi f_z$ and $f_z$ is frequency in the depth with units $m^{-1}$. From this the power spectral density (PSD) is,

$$P(k) = P_0(k)e^{-k^2\sigma^2} \tag{3}$$

where $P_0$ is the PSD of the isotope profile before diffusion and $P$ is the PSD of the signal after diffusion. In real isotope records the measurement process will add some noise $\epsilon$ to the signal,

$$P(k) = P_0(k)e^{-k^2\sigma^2} + \epsilon(k) \tag{4}$$

Therefore it is possible to estimate the diffusion length of a water isotope record by taking its power spectrum and fitting Eq. 4 as a model of the effect of diffusion.

     The white-noise climate assumption of conventional methods relates to the $P_0$ term, prescribing it as frequency independent. 110 A constant $P_0$ means the model contributes all of the power difference between frequencies to diffusion. Also, when water isotope measurements are made of discrete samples, the measurement noise, $\epsilon$, is often considered to be white-noise, as each measurement should be independent of the previous measurement. This does not hold true in a continuous flow analysis system, where the noise can have positive autocorrelation due to mixing within the water lines (Gkinis et al., 2014) which would also influence the shape of the power spectrum. All the data used in this report was discretely measured, so the measurement





noise was considered to be white.

Discrete sampling will also introduce a block averaging effect, further smoothing the data and biasing our diffusion length estimate if not taken into consideration. The magnitude of this smoothing depends on the sampling resolution, and can be computed using the method described in Gkinis et al. (2014). For the 11cm resolution data used here, we compute the corre-

sponding rectangular filter of width 11cm to be equivalent to a diffusion length of ∼3.3 cm. Such small scale smoothing is unlikely to have a significant effect on the heavily diffused deep ice, so the resulting bias is considered negligible.

## 2.5  Fitting methods

There are two common approaches for fitting a spectrum with Eq. 4, a linear regression method applied only to the lower frequencies (Jones et al., 2017) and a non-linear method applied over the entire spectrum (Gkinis et al., 2014; Holme et al.,

2018; Kahle et al., 2018).

The linear approach (Jones et al., 2017) focuses only on the lower frequencies where the power of the climate signal is much greater than the measurement noise term ($P_0(k)e^{-k^2\sigma^2} >> \epsilon(k)$). In this frequency range, Eq. 4 can be approximated as Eq. 3, and by taking the natural log of both sides we get,

$$\ln P(k) = -k^2\sigma^2 + \ln P_0 \qquad (5)$$

Assuming $\ln P_0$ is independent of frequency (Johnsen et al., 2000), we can model $P$ on a logarithmic scale as a function of $k$ with a linear regression which will have a slope proportional to the square of the diffusion length. The cut-off frequency (that is, the upper frequency limit after which the $\epsilon$ term is no longer negligible) is usually manually defined, which is makes this method difficult to generalise and automate. This method relies on the frequency independence of $P_0$, as any changes with

frequency will create a non-linear relationship between $P$ and $k$.

Another possible method of statistically estimating the diffusion length involves modelling $\ln P$ as a function of $f_z$ and fitting all parameters of Eq. 4 including the noise (Gkinis et al., 2014). Unlike the linear approach, this fit can be applied over all available frequencies, removing the subjective step of choosing a cut-off frequency. Variants of this approach exist for

situations where the measurement noise is not white, such as with data measured using continuous flow analysis (Gkinis et al., 2014; Kahle et al., 2018).

While previous studies assumed $P_0$ is equivalent to white-noise, the latter fitting method, unlike the linear one, also allows a frequency dependence of $P_0$, and so it was the preferred choice for our new approach going forwards.



## 2.6 Estimating diffusion length when $P_0$ is not constant

To get a realistic, alternative the model for $P_0$ suitable for longer timescales, we use water isotope data less affected by diffusion from the same site. Assuming the statistics of similar climate states over time are comparable, we estimate $P_0$ empirically from a shallower section of the same ice core spanning a time period of similar length and climate state. The ice diffusion in this shallow record is still negligible, as it is younger, colder and less thinned than the deepest ice. Additionally, any significant firn diffusion is on timescales much shorter than we are analysing. For the MIS 19 case, we take water isotope data from interglacial periods further up in the EDC ice core.

To parameterise the variability across a large range of frequencies we use a power law,

$$P_0(f_t) = \alpha * f_t^{-\beta} \tag{6}$$

which has been shown to be a good approximation of the spectrum of climate variability (Pelletier, 1998; Huybers and Curry, 2006). Here, $\alpha$ and $\beta$ are constants, and $f_t$ is the frequency in the time domain. We work in time units because similar climate variability during different past climate states results in comparable power spectra in the time domain but not in the depth domain, as the annual layer thickness will vary over time and ice depth.

For the case of the MIS 19 record from EDC, an appropriate time period to estimate the climate signal before long-term ice-diffusion is another, more recent interglacial. Using the same EDC ice core, we selected sections of water isotope data from MIS1 (the Holocene), MIS 5 and MIS 9 interglacials, which were retrieved from depths shallow enough to remain unaffected by lower frequency diffusion. Large data gaps are present over the MIS 7 and MIS 11 interglacials, so a reliable power spectral estimate could not be acquired. Interglacial records from deeper in the ice core (MIS 13/15/17) were not suitable for our analysis as diffusion has attenuated the frequencies over which we are inferring the spectrum of $P_0$. Using different interglacial periods to estimate $P_0$ allows us to evaluate the sensitivity of our diffusion length estimate on our choice of interglacial.

## 2.7 Bayesian fitting approach

We used a Bayesian approach for all our power spectrum fitting as it has a several advantages over classical methods. Most importantly, rather than having to either set parameters to specific fixed values, or leave them free to assume any value, the prior distributions of a Bayesian model allow us to put physically realistic restrictions on the values of parameters while also allowing uncertainty in their true values to be propagated through into uncertainty in the final estimate of diffusion length. Additionally, the Bayesian method allows us to specify a gamma distribution which is the true distribution of the errors in power spectral density estimated by Fourier methods (Bloomfield, 2004). The Gaussian approximation assumed by classical fitting in log-space produces a positive bias in the estimated power, which here would affect estimated diffusion lengths.





Using a reduced model without diffusion (Eq. 6 from earlier), we fit our undiffused climate spectra ($P_0$) for MIS 1, MIS 5 and MIS 9 with weakly informative priors of $\alpha \sim$ N(0.1, 1) and $\beta \sim$ N(1.5, 1), giving a large range of possible values to find the best fit for the initial MIS 19 climate signal. The posterior means and standard deviations of $\alpha$ and $\beta$ from these fits were
then used to parameterise informative priors for subsequent fits of the full model including diffusion (Eq. 4) to MIS 19.

$$P(f_z) = \alpha * f_z^{-\beta} e^{-4\pi^2 f_z^2 \sigma^2} + \epsilon \tag{7}$$

For the power of the noise we used an informative prior, $\epsilon \sim$ N(0.07°, 0.02°), as the uncertainty of the d18O data is well defined from the lab based measurements. For the key parameter of interest, the diffusion length, we used a weak prior, $\sigma \sim$ N(0.4°, 0.4°). We set a lower limit of 0 for all parameters in all fits to prevent negative, non-physical, values for $\alpha$, $\sigma$ and
the noise $\epsilon$, and to constrain $\beta$ to power-law red-noise like behaviour. For the error in spectral estimation we used a gamma distribution, $\gamma(\phi, \phi/\hat{P}(f_z))$, where magnitude of the error is proportional to the estimated power. We fit an additional final model with a prior for P0 defined by taking the mean and standard deviation of the $\alpha$ and $\beta$ estimates from the individual interglacials ("Mean" row in Table 2).

The models were defined in the Stan language (Carpenter et al., 2017) and fit using the No-U-Turn sampler (Hoffman et al., 2014) with the R package cmdstanr from Gabry and Češnovar (2021). For each fit, we sampled 2000 values from 4 independent chains, with the first 1000 iterations discarded as warm up. Inspection of Rhat values and traceplots of the posterior model parameters indicated that chains were well mixed for all parameters and all models converged.

### 2.8 Conventionally estimated EDC diffusion lengths

In order to determine the significance of this new method, we first estimate the diffusion length using the conventional model where the climate variability $P_0$ is considered constant across all frequencies, the second method in Section 2.5. However, we still use our Bayesian approach to circumvent the bias of the least squares estimator. We use the same $\sigma$ and noise priors as before, and a weak $\alpha$ prior of $\alpha \sim$ N(0.5°, 0.1°). We apply this to a running window of 500 data points across the full high resolution EDC record.

## 3 Results

### 3.1 New $P_0$ estimate

The full power spectra of the selected interglacial time periods used to constrain $P_0$ proved difficult for a direct power-law fit. Different timescales can have different power laws (Pelletier, 1998) and this is evident in the power spectra, which seem to level off at frequencies above $3kyr^{-1}$ (Fig. 2). Considering also that the lowest few frequencies in the spectra are biased
due to tapering, we did not apply the $P_0$ fit over the whole spectra. We heuristically selected a frequency range between $0.25kyr^{-1} < f_t < 2.5kyr^{-1}$ which can be reasonably modelled with a power law for each interglacial. We also devised a





technical approach to compute this range, taking care to include all frequencies relevant for the MIS 19 diffusion length estimate, as explained in Appendix A. This closely matched our manually chosen range, which we proceeded with for simplicity.

Using the Bayesian sampling method, the power-law fit is applied to each of the three more recent interglacials within these frequency bounds, with best fits using the mean $\alpha$ and $\beta$ values per interglacial shown in Fig. 2. The corresponding average parameter values are shown in Table 2, with errors representing the standard deviation of the Bayesian estimates. We also use the mean of both parameters across the three interglacials fits to represent the average interglacial. This gave Gaussian prior distributions of $\alpha = 0.0176 \pm 0.0025$ and $\beta = 1.19 \pm 0.28$, with the strength taken as the standard deviation of the interglacial

parameter means, accounting for differences between the statistics of the similar climate states.

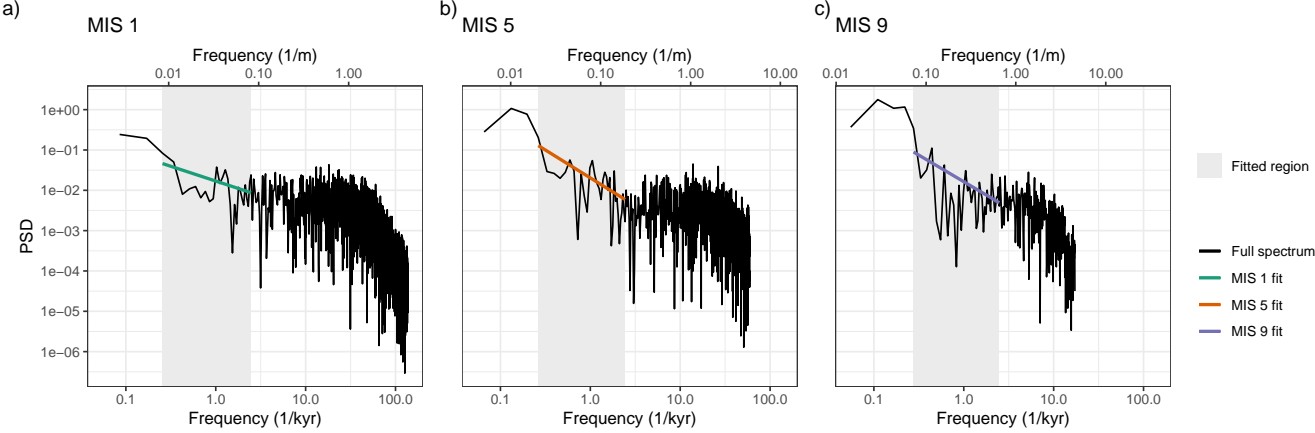

**Figure 2.** Best power-law fit for the power spectra of (a) MIS 1, (b) MIS 5 and (c) MIS 9, over the frequency range $0.25 kyr^{-1} < f_t < 2.5 kyr^{-1}$ (shaded grey). The effect of firn diffusion can be seen attenuating the highest frequencies, which were excluded from the power-law fit.

**Table 2.** Alpha and beta values estimated from each interglacial fit in Fig 2, and their mean. Errors in the interglacial parameters are the standard deviation of the sampled values, while the error in the mean value is the standard deviation between the average interglacial values.

| Interglacial | $\alpha(‰^2 m)$ | $\beta$ |
| --- | --- | --- |
| MIS 1 | $0.0160 \pm 0.0032$ | $0.86 \pm 0.27$ |
| MIS 5 | $0.0205 \pm 0.0039$ | $1.38 \pm 0.31$ |
| MIS 9 | $0.0163 \pm 0.0030$ | $1.31 \pm 0.23$ |
| Mean | $0.0176 \pm 0.0025$ | $1.19 \pm 0.28$ |





## 3.2 Improved diffusion length estimate for MIS 19

While prior distributions of $\alpha$ and $\beta$ could be derived from data, we still need to choose priors for the measurement noise and the diffusion length itself for our MIS 19 diffusion length estimate. A Gaussian prior distribution of $0.07 \pm 0.02‰$ is sufficient for the $\epsilon$ term as the uncertainty of the $\delta^{18}O$ data is well defined from the lab based measurements. The diffusion length, $\sigma$, was

given a very weak Gaussian prior distribution of $0.4 \pm 0.4m$, allowing the model almost complete flexibility to find the best fit. Using the mean $\alpha$ and $\beta$ priors, a diffusion length of $31 \pm 5cm$ (95% confidence) was estimated. A full table of the results for each interglacial is shown in Table 3.

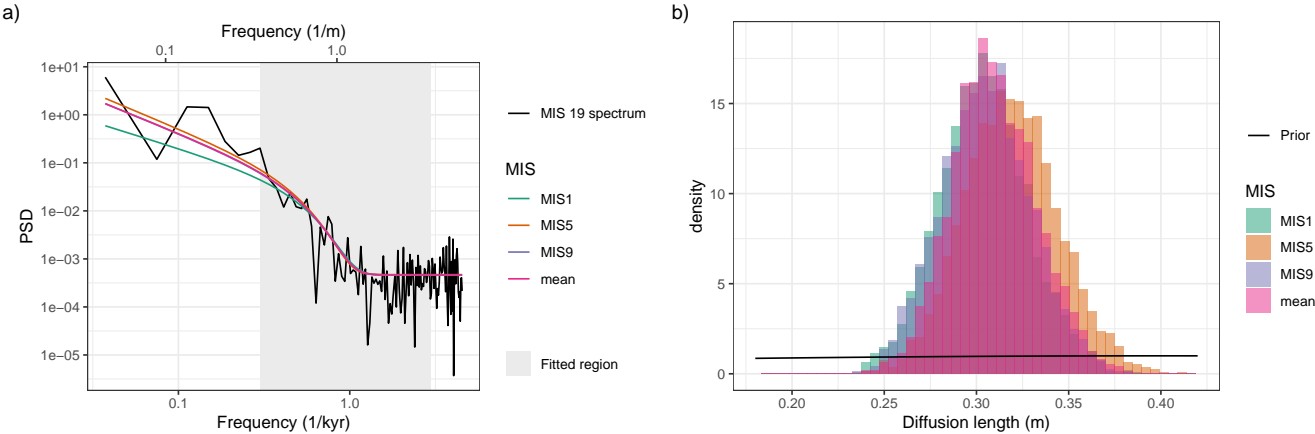

**Figure 3.** (a) Best fit of the power spectrum of MIS 19 for each $P_0$ prior defined using the younger interglacials and the mean result. The fit was only applied over the shaded region, matching the shaded region used in the power-law fits 2 (b) Histograms of Bayesian posterior sampled diffusion length estimates for all four scenarios. The weak diffusion length prior is shown in black.

**Table 3.** All parameters estimated from the MIS 19 spectrum, using different $P_0$ ($\alpha$ and $\beta$) priors. Uncertainties represent 95% confidence intervals.

| Interglacial | $\alpha(‰^2 m)$ | $\beta$ | $\sigma$ (cm) | $\frac{\sqrt{\epsilon}}{dz}$ (Noise (‰)) |
|---|---|---|---|---|
| MIS 1 | $0.015 \pm 0.005$ | $1.2 \pm 0.4$ | $30 \pm 5$ | $0.0653 \pm 0.0008$ |
| MIS 5 | $0.017 \pm 0.006$ | $1.5 \pm 0.5$ | $32 \pm 5$ | $0.0659 \pm 0.0009$ |
| MIS 9 | $0.015 \pm 0.005$ | $1.5 \pm 0.4$ | $31 \pm 5$ | $0.0656 \pm 0.0009$ |
| Mean | $0.015 \pm 0.004$ | $1.4 \pm 0.4$ | $31 \pm 5$ | $0.0655 \pm 0.0009$ |

## 3.3 Comparison of conventional and new approach

Estimating diffusion length using an incorrectly parameterised best fit model can significantly bias the result. To demonstrate

this, we use the conventional method in Section 2.8 and apply it to the theoretical spectrum of a diffused power law with measurement noise, with $\alpha = 1$, $\beta = 1.5$, $\sigma = 0.3$ and $\epsilon = 0.07$. The computed best fit greatly misrepresents the lower frequencies





(Fig 4) and estimates a diffusion length of 0.415, an overestimation of almost 40%.

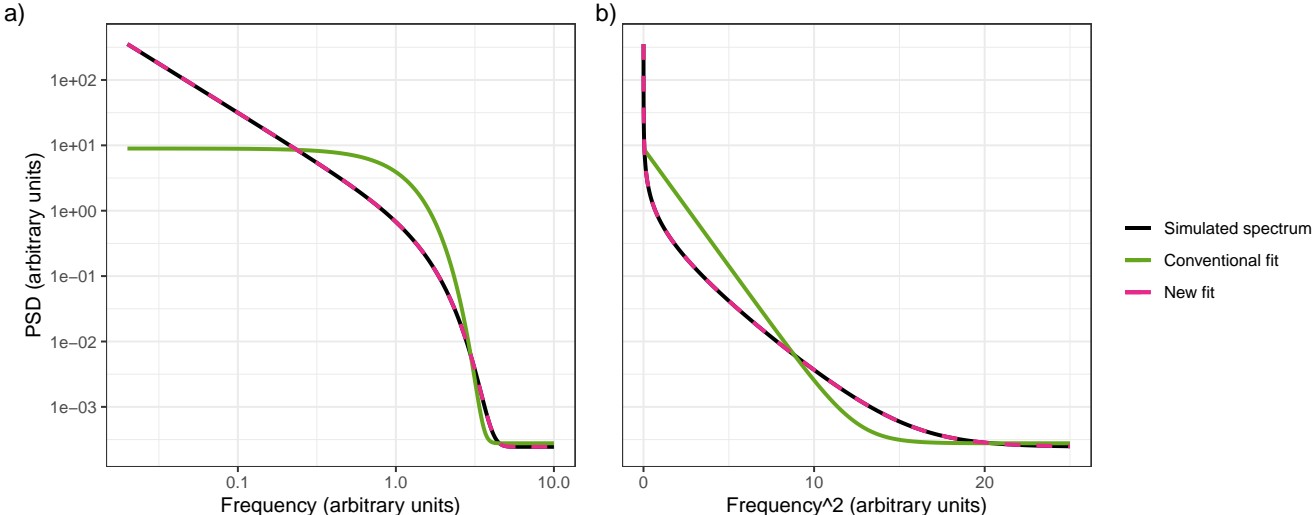

**Figure 4.** Best fits of a theoretical power spectrum of a deep ice-core record simulated using Eq. 4, plotted on (a) log-log and (b) log vs $f^2$ axes. The latter demonstrates how the assumed linearity of the low frequencies in the first fitting method from Section 2.5 breaks down when some underlying power-law behaviour is present.

Aware of this bias, we applied the conventional method to the MIS 19 spectrum to evaluate the difference between the two
estimates, using the Bayesian approach with a very weak $P_0$ prior of $1 \pm 10\ \%o^2$m and the same $\sigma$ and noise priors as before. Directly comparing the two fits, our new method demonstrates a clear qualitative improvement (Fig. 5a). The conventional approach estimated a diffusion length of $55 \pm 6$ cm, over 77% larger than the new method. Adding our new result to the conventionally estimated Dome C diffusion length record (Fig. 5b) gives a new impression on the scale of diffusion in the deepest section of Dome C.





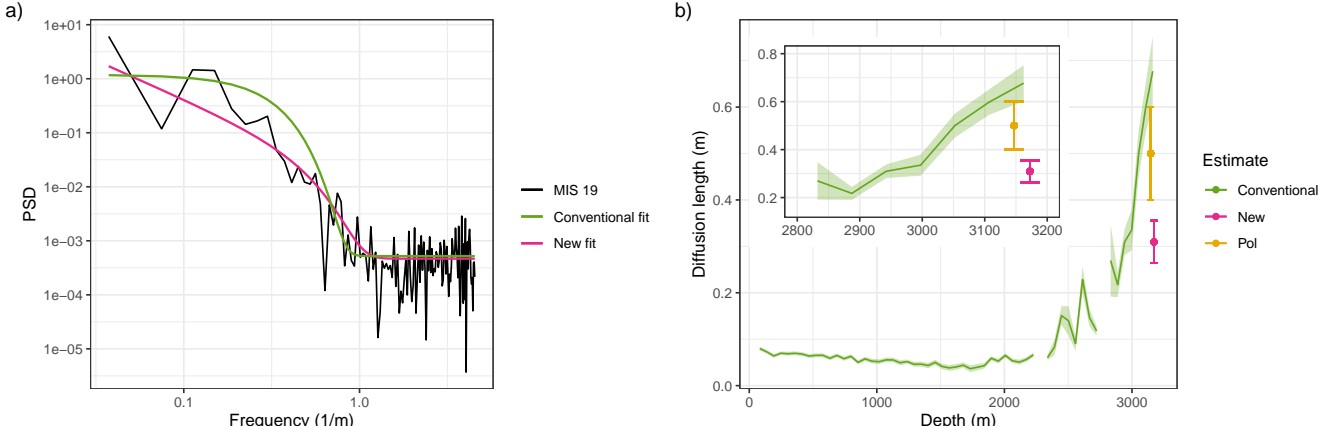

**Figure 5. (a)** Comparison of MIS 19 power spectrum best fits using the conventional, white-noise method (green) and the new power-law method (pink). **(b)** Conventional diffusion length profile of EDC record (green) and new diffusion length estimate (pink). MIS 19 diffusion length estimate from Pol et al. (2010) is also shown for comparison (yellow). **(inset)** Zoom in on the deepest section of the EDC record.

## 4 Discussion

### 4.1 Overcoming the estimation bias of the conventional method in deep ice

Applying the conventional method to a running window over EDC, the estimated diffusion lengths rapidly increase from 20 cm at a depth of 2800 m to nearly 60 cm at the bottom of the ice core, a depth of almost 3200 m (Fig. 5b). The average 500 data point window in this depth range covers 50 kyr, a timescale over which variability cannot be considered uniform across all frequencies. Furthermore, this interval in the record spans four glacial-interglacial cycles, with some windows containing the transition between these states, contributing to more low frequency variability. This is inconsistent with the common assumption of approximating the original, undiffused record as white-noise, strongly biasing conventional estimates of deep ice diffusion. Adding frequency dependence to $P_0$ in the estimator model through prescribing power-law scaling helps to eliminate such biases.

### 4.2 Isotopic variability before diffusion and interglacial assumption

The downside of the proposed method is that it needs information on the isotope variability before diffusion ($P_0$). In our estimate of this undiffused isotopic signal, we assumed the spectral properties of more recent interglacial periods are comparable to that of the MIS 19 interglacial. It could be argued this is a logical leap, as the interglacials vary in duration and amplitude. However, our diffusion length estimate was insensitive to the interglacial chosen (Fig. 3b, Table 3). For our final result estimated using the mean of the $P_0$ interglacial fits, the strength of the priors of $\alpha$ and $\beta$ are taken as the standard deviation across the individual interglacial fits, corresponding to the uncertainty of our average interglacial. Regardless of whether these precautions represent the full uncertainty in our assumption, it is certainly a more accurate representation of isotopic variabil-



ity over multi-millennial timescales than white noise. It is usually assumed that isotopes in ice cores are faithful recorders of multi-centennial and longer temperature variability. Therefore, $P_0$ is dominated by the climate variability on these time-scales.
This would also allow for the use of other proxy information, for example from marine sediment cores (Shakun et al., 2015), in future studies to test how appropriate it is to use $P_0$ from more recent interglacials to approximate the variability in MIS 19.

### 4.3   Advantages of the Bayesian approach

Our implementation of a Bayesian fitting approach enables our prior knowledge of the physically realistic values of the parameters to be incorporated into the fit (Gelman et al., 2013). Using a fixed power-law slope to represent our estimation of the
past climate statistical properties would be restrictive, and its accuracy would influence our diffusion length estimate. On the other hand, allowing complete freedom of the $P_0$ spectrum fit could produce a biased diffusion length estimate as the shape of $P_0$ may account for some of the loss of variability due to diffusion. The Bayesian method offers a middle ground where the fit parameters are suggested through priors, and the strength of the suggestion depends on the confidence of our knowledge of the parameter. Given our inference for $P_0$ is an assumption, this is ideal as it allows some flexibility during the fitting process
to account for differences between the MIS 19 interglacial and the more recent interglacials, without overcompensating for diffusive attenuation effects. In other words, the priors enabled our knowledge and confidence in the parameter values to be incorporated into the fitting process, allowing the uncertainty of the parameters to propagate through to the fit. A further advantage of this method was its capability to correctly treat the residuals of the model as a gamma distribution. Standard non-linear fitting methods assume a Gaussian distribution of the residuals, but this is not the case for power spectral density estimators.

### 4.4   Implications of the new diffusion length estimate for MIS 19

Our estimate of $31 \pm 5 cm$ for the diffusion length over the MIS 19 interglacial period, relative to earlier estimates of 40 cm - 60 cm (Pol et al., 2010), has important implications for the interpretation of this part of the EDC ice core. This is because the remaining power of certain frequencies after diffusion is extremely sensitive to the diffusion length. In the EDC record, millennial variability would be reduced to around 1% of the record before diffusion for our new estimate, compared to 1/100,000
for the conventionally estimated $55 \pm 6 cm$. The former offers the opportunity to reconstruct millennial variability using deconvolution, whereas this is not possible for the 55 cm case given the magnitude of measurement noise. The significant decrease in our diffusion length estimate versus conventional estimates can be explained by the introduction of a 'red-noise' behaviour from our $P_0$ fit 4. Red-noise contains more variability in low frequencies than high frequencies, and so may account for some of the power difference across frequencies which would otherwise be attributed entirely to diffusion, as in the conventional
case. The new estimate is much closer to the diffusion lengths at the bottom of the EDC ice core modelled from physical first principles, which average around 20 cm (Pol et al., 2010; Grisart et al., 2022), reducing the discrepancy between the empirical and analytical results. This new result offers a much more optimistic outlook for recovering millennial-scale climate features from future deep ice-core projects such as the BE-OIC.



## 4.5 Future outlook

Possible improvements to the precision of our estimate could be made with higher sampling resolution or data spanning longer time periods. Increasing the resolution would provide some opportunity to reduce the uncertainty, but the effect will be limited as the high frequencies do not contribute to the diffusion length estimate, with the main improvement arising from the effective measurement noise reduction at a given frequency. It would also lessen the smoothing effect of block averaging due to the rectangular sampling scheme, although in the future this could be directly incorporated into the fit. Alternatively, increasing the time-span of our selected water isotope record would improve the reliability of our $P_0$ estimate. However, for MIS 19 the record could not be extended without including data from glacial periods, different climate states with different variability structures, which would complicate our $P_0$ model.

While the focus of this new method was to improve the diffusion length estimate for deep ice, it should also be considered for application to shallower ice cores. power-law behaviour is observable in the climate across multi-millennial to sub-annual timescales (Pelletier, 1998), which suggests younger ice cores and firn cores could also benefit from this modified approach. Specifically at core sites with higher accumulation rates, and therefore less stratigraphic noise, such as the WAIS Divide ice core (Jones et al., 2018), a power law $P_0$ may offer a more accurate estimation than white-noise.

Perhaps the main disadvantage of this new method compared to conventional estimation methods is the necessary inference of the undiffused isotopic profile from elsewhere in the ice core. It would be much more practical if it were possible to generalise the approach for an entire ice core, acquiring a full diffusion length profile with depth. This is potentially achievable by using a different parametrisation for $P_0$ that is valid across a broader frequency range. Using white noise (reflecting depositional noise) superposed on a piecewise power-law scaling (reflecting internal climate variability vs. multimillenial orbitally forced variability) might be one option. Another idea is to test if non-diffused parameters such as the dust content from the same section of the ice core can be related to the climate variability and thus P0.

## 5 Conclusions

We have described a new approach for water isotope diffusion length estimations in deep ice cores, resolving the biased assumption of a white-noise undiffused climate signal. Our method instead implements a power-law slope inferred from water isotope sections of a similar climate state in the shallower parts of the ice core, better representing the climate on millennial scales. Incorporating Bayesian statistics enables us to use priors, chosen based on our knowledge of the parameters and propagating our uncertainties into the fitting procedure. We applied our new method to the MIS 19 water isotope record from the bottom of the EDC ice core, estimating a diffusion length of $31 \pm 5$ cm, a 23% reduction on the smallest previously estimated value of 40 cm (Pol et al., 2010). A smaller diffusion length offers a more optimistic outlook for the preservation of millennial climate signals in oldest ice projects such as the BE-OIC. Future work could be made to generalise the process for entire





ice-core records, possibly through adjusting the Bayesian priors for different climate states or, alternatively, inferring $P_0$ from non-diffused parameters.

*Code availability.* Code used in this study is available upon request

*Data availability.* The data used in this study was previously published and can be found at: https://doi.org/10.1594/PANGAEA.939445

**Appendix A: Defining the frequency range of the fit**

The $P_0$ power-law fit need only be applied over the frequencies most sensitive to small changes to the diffusion length, as lower frequencies are unaffected by diffusion and higher frequency variability is attenuated to such an extent that measurement noise dominates. Therefore, an estimate is only necessary over the frequencies in between these two extremes. To find the relevant frequency range, we rearrange Eq. 3 for frequency,

$$f_t = \frac{\sqrt{-\ln\left(P(f_t)/P_0(f_t)\right)}}{2\pi\sigma}\bar{\lambda} \tag{A1}$$

where $\bar{\lambda}$ represents the mean annual layer thickness and is required to convert the diffusion length into the time domain. Here, the term $P(f_t)/P_0(f_t)$ represents the fraction of power remaining after diffusion at the frequency $f_t$. Therefore, we can calculate a frequency range for the fit by defining an upper and lower limit of power loss, assuming an approximate diffusion length. Given we do not know the diffusion length, we take a conservative guess to prevent excluding important frequencies.
For the MIS 19 case, we approximate a minimum diffusion length of 15 cm and a maximum diffusion length of 60 cm, and use these values to estimate the upper and lower frequency limit respectively. The lower limit is defined as the frequency where the remaining power first drops below $1/e$, while the upper limit is defined as the frequency after which the remaining power is below the measurement noise level. For the latter definition, the power of the measurement noise can be reliably estimated from flat of the diffused power spectrum, while the assumed $P_0$ is estimated from the highest frequencies in the power spectra
(giving a conservative, larger frequency range as this is a larger $P_0$ than would be expected for the higher frequencies).

The measurement noise was estimated from the MIS 19 power spectrum, which tails off over the highest frequencies at a PSD of $5.4 \times 10^{-4}\text{‰}^2 m$, corresponding to a measurement noise of $\pm 0.07\text{‰}$. Then, given $\bar{\lambda} = 8.4 \times 10^{-4} my^{-1}$ or $0.84 mkyr^{-1}$ and assuming an upper diffusion length estimate of 60 cm (Pol et al., 2010), a power drop to $1/e$ occurs at a frequency of
$f_t = 0.22 kyr^{-1}$. Likewise, using the mean annual layer thickness and a lower diffusion length estimate of 15 cm from models (Pol et al., 2010), the power drops below 10% of the estimated measurement noise at $f_t = 2.45 kyr^{-1}$.



*Author contributions.* TL and FS designed the study. TK and TL contributed to the spectral analysis and signal processing. AD contributed to the Bayesian estimation. VG provided insights on diffusion and diffusion length estimation. FS performed the analysis and wrote the manuscript. All authors contributed to the interpretation and to the preparation of the final manuscript.

*Competing interests.* The authors declare that they have no conflict of interest.

*Acknowledgements.* This project has received funding from the European Union's Horizon 2020 research and innovation programme under the Marie Sklodowska-Curie grant agreement No. 955750. This project has received funding from the European Research Council (ERC) under the European Union's Horizon 2020 research and innovation programme (grant agreement No. 716092). This project has also received funding from The Villum Foundation (The whisper of ancient air bubbles in polar ice, 00028061 and Unraveling paleo-climate knots with lasers, 00022995).



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
