# Peer review of "Novel approach to estimate the water isotope diffusion length in deep ice cores with an application to MIS 19 in the EPICA Dome C ice core"

_EGUsphere, 2023_

## Author Comment (AC1)

**Response to comments from Reviewer 1**

Authors' replies in blue. All reference to figures, equations or lines in the paper refer to the initially submitted paper.

The paper presents a novel technique that aims to solve an intrinsic challenge existent when unravelling paleo-climate data from deep ice cores. The paper is well-written and structured, the research is original, and the conclusion provides a promising outlook in the journey towards restoring paleo-climate data from the oldest ice. I therefore recommend this paper for publication in The Cryosphere after some minor revisions have been addressed.

My primary concern is that it is unclear to me why the method described in Sec. 3.1 returns the P0(f) relation that can be used in deep ice diffusion estimates. As the authors also write on line 103, P0(f) is the PSD of the isotopic profile *before* diffusion. They then proceed to use ice core sections with ages more than 10 kyr which at a minimum have been subjected to firn diffusion which must have altered the initial signal. They argue that the time horizon that they assess is unaffected by firn diffusion, but firn diffusion has completed its alteration of a deposited snow layer within 70-200 years (Johnsen et al., 2000). So, I recommend the authors to extend their argumentation to include why firn diffusion is negligible/irrelevant to their P0(f) estimation methodology.

We thank the reviewer for their positive feedback and recommendation for publication. Their first concern focuses on the significance of firn diffusion in the power spectra we use to estimate our undiffused climate signal $P_0(f)$. We apply our $P_0(f)$ fit on the three interglacials MIS 1, MIS 5 and MIS 9 which all span time periods of 10k+ years. As the reviewer correctly pointed out, all three ice core sections will have undergone firn diffusion during the initial densification process. However, as we show below, the frequencies significantly affected by firn diffusion are much higher than the frequencies used to estimate $P_0(f)$ and therefore firn diffusion does not affect our $P_0(f)$ estimates.

For the Dome C site, firn diffusion lengths reach a maximum of 8 cm at a depth 50 m (Johnsen et al., 2000), before reducing due to layer thinning and the fact that the firn diffusive process ceases below pore close-off. Any ice diffusion is negligible on these timescales and temperatures.

The upper frequency limit of our $P_0$ fit is 2.5 kya$^{-1}$ (the right side of the shaded regions in Fig. 2), or a timescale of 400 years. This timescale spans at least 11 m in the firn diffusion regime, a scale unaffected by a diffusion length of only 8 cm. To show this we can rearrange Eqn. 3 to estimate the remaining fraction of power for a given frequency and diffusion length:

$$\frac{P(k)}{P_0(k)} = e^{-k^2\sigma^2} = e^{-(2\pi f)^2\sigma^2} = e^{-(2\pi*\frac{1}{11})^2*0.08^2} = 0.998$$

So the highest frequency has lost only 0.2% of the power due to firn diffusion, which is negligible in the power spectra. We will add the following comment at Line 209 of the paper to address this:

"This frequency range also excluded the significantly diffused higher frequencies, with the highest included frequency having lost only 0.2% of its initial power due to firn diffusion, which makes a negligible difference in the power-law fit."

It is also unclear to me whether they calculate the diffusion lengths estimate on a time or depth basis. The text and figures indicate it is on a time domain but the tables and presented values are depth-domain estimates. Moreover, if the authors take a time-domain approach, then there are some complications that they have not covered. For instance, there is an inherent uncertainty associated with an ice core chronology that is not accounted for in time scale estimates. This makes time-domain diffusion length estimates more uncertain that depth-domain estimates. This can be accounted for in the PSD estimation, and I would expect that the Bayesian framework that the authors are adopting are suitable to handle such uncertainties. So, I'd encourage the authors to (1) clarify what types of diffusion length estimates that they are calculating, and (2) consider implementing chronology uncertainties in the methodology in case they estimate time-dependent P0(f) and diffusion length estimates, and (3) that they specify the conversion factors from time-dependent diffusion length estimates to the depth-dependent estimates that are presented in the table.

All estimates of diffusion length in the manuscript are made and provided on a depth basis, and thus the data is equidistant. All $P_0$ slope estimates are done in the time domain, as they characterise a climatic input signal that should be largely stationary in time, but not in depth (as the same climatic isotope signal will be on different depth frequencies when the layer thickness changes). For these estimates, we use time frequency (1/ky), by linearly interpolating the $\delta^{18}O$ values onto an equidistant age scale. The timeseries selected for $P_0$ fits also contain some missing data, which is resolved during the linear interpolation process. Both of these factors introduce some error to the power spectra, but as we will demonstrate below, the effect on the final diffusion length estimate is negligible.

To see the effect this might have on the resulting $P_0$ fits, for each interglacial (MIS 1/5/9) we simulate a very high resolution time series with the same parameters ($\alpha$ and $\beta$ values) as the interglacial. We then bin the data to the unevenly spaced axis of the interglacial, and remove data where the interglacial data has gaps. We

compare the spectra and fits with evenly binned data with no gaps, representing the ideal, unbiased spectra. The results for each interglacial are shown below in Fig. R1.

[Figure]

**Fig. R1:** Effect of linear interpolation on three simulated power spectra with parameters equal to the 3 selected interglacial records: a) MIS 1, b) MIS 5, c) MIS 9. The resulting fits still fall within our confidence intervals.

We found the linear interpolation has a negligible effect in the frequency range the fit is applied over, as it only introduces error in much higher frequencies. For MIS 1 and MIS 9 there is not much data missing, so the interpolated fit matches almost perfectly with the simulated complete timeseries. For MIS 5, while enough data is missing that the fit deviates slightly from the ideal case, the resulting fit falls within the 90% confidence interval. Given it only contributes partially to a suggestive prior then the MIS 19 fit is not strongly affected.

Once the $P_0$ slopes are estimated, the alpha values are converted to depth frequency for the MIS 19 fits, as seen in Table 2. The conversion is done by multiplying by the mean annual layer thickness across the MIS 19 section, which will also introduce some error. To evaluate the significance of this error, we used the maximum and minimum annual layer thickness across the MIS 19 section as our conversion factor. In both scenarios the diffusion length estimate differed by less than 1 cm from the mean annual layer thickness scenario, well within our uncertainty.

The above plots and explanation will be added in an Appendix section and the following comment will be added to Line 84.

"The effect on the spectra was small, and did not significantly impact the final result (see Appendix)."

Finally, I suggest that the authors define what types of diffusion lengths that they refer to throughout the paper. Are they PSD-estimated, firn diffusion lengths, ice-equivalent diffusion lengths, etc.? There are occasionally references to modelling

output estimates or other studies (e.g., lines 49-51), and it will help the reader to ensure that the same metric is being used when they compare magnitudes of diffusion.

We recognise the referenced diffusion lengths from previous studies are sometimes unclear regarding whether they are empirically estimated using the PSD or physically/numerically modelled. We will edit such references to clarify.

**Minor Comments**

· Lines 178 – 189: I'd like an extended and visual assessment of how sensitive the diffusion length outputs are to changes in priors. Given your weakly defined priors, it seems to me that it isn't that sensitive, but I think it would be valuable to emphasize this further given this is a new methodology. This could be in an appendix. Moreover, it would be valuable with some suggestions or guidelines for prior values to select for different climate regions like East Antarctica, West Antarctica and Greenland.

Below are plots showing the prior and posterior distributions for both the $P_0$ fits (Fig. R2) and the mean MIS 19 fit (Fig. R3). Both the priors for the $P_0$ fits and the priors for sigma and noise in the MIS 19 fit are uninformative, made clear by their wide spread relative to the widths of the posterior distributions. The resulting posterior distributions are therefore insensitive to our choice of prior. For the mean MIS 19 fit, the alpha and beta priors are much more informative, which is desired as they are derived from the $P_0$ fits. These figures and above explanation will be added as an appendix to the paper.

[Figure]

**Fig. R2:** Posterior distributions for the three $P_0$ fits and their corresponding uninformative priors.

[Figure]

**Fig. R3:** Posterior distributions for the mean MIS 19 fit and their corresponding priors. The informative priors for alpha and beta values are acquired from the $P_0$ fits. Both diffusion length and noise priors are uninformative.

Given the insensitivity of the fit to our weak priors, similar prior values can be used regardless of climate region (e.g. wide Gaussian spreads). Diffusion length models using physical parameters could be used as a prior for other sites, but here we wanted to keep the result as independent as possible from the model. Only the alpha and beta values (the $P_0$ slope) need an informative prior, for which appropriate values can be derived from less diffused water isotope data from similar climate states further up in the ice core, as we have done in this study.

· Line 177 - please elaborate a bit on what N(0,02,0.07) means in terms of the gamma distribution. For instance, can you specify what N refers to in this case. Is it a normal, uniform or gamma distribution? I could assume the N(0.1, 1) and N(1.5, 1) refers to the shape and scale coefficients of the gamma distribution but please state it explicitly then.

The following explanations will be added in the text:

At Line 178: "Here, N(x, y) refers to a Normal distribution of mean 'x' and standard deviation 'y'."

At Line 185: "For the error in spectral estimation we used a gamma distribution, $\gamma(\varphi, \varphi/\widehat{P}(f_z))$, where φ represents the scale parameter and $\varphi/\widehat{P}(f_z)$ represents the shape parameter."

· Line 177, you write that your fit undiffused climate spectra, but are these spectra really undiffused? I agree that they have been subjected to less diffusion that MIS 19 but I think this is something that should be addressed. See main comment.

As stated in the response to the main comment, and now will also be stated in the revised manuscript, while the spectra have been diffused on the shortest timescales, they are unaffected within the range we apply our $P_0$ fit.

· Line 190, as this is a seminal paper on a novel approach, then I would like to see the underlying figures in an appendix such that it is clear to the reader how the model converges from the a pr*iori* guess towards the underlying distributions. This will be helpful to future users when they are deploying your framework in practice.

Below (Fig. R4) is a trace plot of estimated diffusion length values for the mean $P_0$ MIS 19 fit. All 4 chains have different initial values but quickly converge to the same estimated value during the warm-up phase (indicated by the shaded region). The warm-up values are discarded in the final estimate. Similar plots were found for all other parameters.

[Figure]

**Fig. R4:** Trace plot of sampled diffusion length values for the 'mean' fit.

· Figure 2. The power-law estimates don't seem to fit the signal within the grey-shaded area that well (particularly fig. 2c for MIS 9). Why is that? The coefficients' standard deviations seem small relative to the visual deviation, so perhaps you could update Figure 2 with the confidence intervals from the estimated parameters?

Since the fits are applied on a log-log plot, there are more data at the high frequencies than the lower frequencies, resulting in what looks like a skewed fit, especially for MIS 9. We have updated the figure to include 90% confidence intervals as requested.

· Figure 5. Missing reference in Figure 5 caption

We have added a reference in the figure caption to Johnsen et al. (2000) which is the method used in Pol et al. (2010). It was perhaps unclear that we estimated the full conventional diffusion length profile seen in Fig. 5b, so we have clarified this in the text.

---

## Author Comment (AC2)

**Response to comments from Reviewer 2**

Authors' replies in blue. All reference to figures, equations or lines in the paper refer to the initially submitted paper.

The authors use a novel approach to estimate the strength of the molecular diffusion in the ice dated back to MIS 19, the oldest interglacial recorded in an Antarctic ice core. The diffusion length is deduced from the analysis of the spectral properties of the isotopic time series with the use of a Bayesian approach. The only, but important, assumption is that the spectral properties of MIS 19 are similar as those of younger isotopic stages (MIS 1, MIS 5 and MIS 9).

The authors conclude that the diffusion length is 31 +/- 5 cm, which is much shorter than in the previous works (40-60 cm). It's a very good news for the climatologists seeking for the Earth's oldest ice, and this is one of the reasons why I like this manuscript.

We are glad the reviewer recognises the improvement of our new approach and the optimistic outlook it provides for current and future deep ice core studies

I do not have major comments on the MS, only a couple of minor suggestions:

Line 61 – you have a citation of this dataset in the "Data availability" section, so it's possible to remove it from here.

Removed in the revised version

Table 1 – the depth range for MIS 1 starts from 7.755 m and the time range from 0 ka, but at the depth of 7.6 m the age a priori cannot be 0 years.

This happened due to rounding, corrected in the revised version

Line 87 – significantly affect?

Corrected in the revised version